# cMoLLM at Scale: Horizontal Scaling Laws for Convolutionally-Gated Mixture-of-LLMs

Xin Yang [1]  Yemin Wang [2]  Mingda Liu [3]  Letian Li [4]  Shuaishuai Cao [5]  Zhengxiao He [6]  Ryan Dong [7] [†]

## Abstract

Scaling large language models (LLMs) has driven their success, yet dense Transformers couple capacity and computation: every parameter is activated for every token, making training and inference costs grow linearly with model size—a critical bottleneck as models approach trillion-parameter regimes. We aim to scale capacity through MoE-style mixture throughout the LLM pipeline rather than only the FFN. Prior pipeline-level approaches include ParaScale(Chen et al., 2025), which introduces virtual tokens and parallel streams but incurs substantial overhead and suffers from homogenized routing and gradient collapse, and AltUp(Baykal et al., 2023), which uses an auxiliary prediction branch but offers limited adaptivity and slow convergence. We establish that MoE-style mixture layers can be reformulated as variable-kernel dynamic convolutions, where each expert corresponds to a $1\times1$ convolutional kernel and routing implements input-conditioned kernel aggregation. Building on this equivalence, we introduce cMoLLM: a convolutionally gated mixture-of-LLMs that routes over end-to-end streams through fully differentiable dynamic convolution. In GPT-2-style models trained on FineWeb, cMoLLM improves language modeling perplexity and downstream GLUE and SQuAD accuracy under matched compute, with better stream utilization, more stable optimization, and favorable scaling compared to ParaScale- and AltUp-style baselines.

---

[1]School of Mathematical Sciences, Zhejiang University [2]Xiamen University [3]Institute of Science Tokyo [4]Shenzhen International Graduate School, Tsinghua University [5]School of Central South University [6]Department of Electrical and Computer Engineering, University of Arizona [7]Independent Researcher. Correspondence to: Ryan Dong <zwdong618@gmail.com>.

*Proceedings of the $43^{rd}$ International Conference on Machine Learning*, Seoul, South Korea. PMLR 306, 2026. Copyright 2026 by the author(s).

## 1. Introduction

The remarkable success of large language models (LLMs) (OpenAI, 2025; Anthropic, 2025; Yang et al., 2025) is closely tied to scale: increasing model capacity consistently improves performance across diverse tasks (Radford et al., 2019; Brown et al., 2020; Kaplan et al., 2020; Hoffmann et al., 2022; Yang et al., 2026; Cheng et al., 2025). Training and inference cost grow roughly linearly with model size, as every parameter is activated for every token. As models approach trillion-parameter regimes, this linear compute–capacity coupling has become a dominant bottleneck, motivating the search for more parameter-efficient ways to expand capacity.

A natural direction is to use a Mixture-of-Experts (MoE)–style approach to scale capacity: route tokens across multiple sub-models and combine their outputs. Most prior MoE work applies this only to the **feed-forward network (FFN)**: experts are FFN blocks, and routing is typically discrete (e.g., Top-$K$), which leads to expert collapse, skewed utilization, and brittle training (Shazeer et al., 2017; Fedus et al., 2022). We instead pursue pipeline-level mixture: routing across entire LLM pipelines (or stream-wise sub-models) rather than FFN-only experts.

Two works are directly relevant. **AltUp** (Baykal et al., 2023) widens token representations and uses a virtual prediction branch to update inactive blocks; it increases effective capacity with small parameter overhead but relies on a fixed, hand-designed structure that is not fully adaptive and often converges slowly. This increases sequence length and thus compute; moreover, routing can homogenize across streams, leading to gradient collapse and training instability. We seek a new pipeline-level scaling mechanism that keeps the MoE-style routing across sub-models intuition, avoids virtual tokens, auxiliary prediction heads, and the pitfalls above.

Our starting point is a theoretical insight: MoE-style mixture layers can be exactly rewritten as dynamic convolutions with input-dependent (variable) kernels. Formally, each expert corresponds to a $1\times1$ convolutional kernel; the router performs input-dependent mixing of these kernels (Theorem 4.1). This establishes MoE–dynamic convolution

*Figure 1.* **Teaser of cMoLLM.** cMoLLM scales capacity at the pipeline level via convolutionally-gated mixture over end-to-end streams, yielding better perplexity and downstream accuracy than dense baselines under matched compute, without virtual tokens or auxiliary heads.

*Table 1.* Systematic qualitative comparison of pipeline-level capacity scaling methods: ParaScale, AltUp, and cMoLLM.

| Method | Extra tok. | Aux. branch | Routing | Compute | Stability |
|---|---|---|---|---|---|
| ParaScale | Yes | No | Parallel streams | Higher | Collapse risk |
| AltUp | No | Yes | Fixed pred. branch | Low | Slow convergence |
| cMoLLM | No | No | Dynamic conv. | $\sim$dense | Stable |

equivalence and gives a unified lens for analyzing sparse, conditional computation. We then approximate this ideal: each "stream" uses a distinct convolutional kernel, and we mix streams via soft, fully differentiable gating—no Top-$K$, no low-rank factorization.

Building on this, we propose **cMoLLM**: a convolutionally-gated mixture-of-LLMs that applies conditional computation to the entire Transformer pipeline (Figure 1). We maintain a small set of end-to-end "streams," each associated with its own $1{\times}1$ kernel; a lightweight gating network produces input-dependent mixture weights, and the mixed kernel is applied via standard (grouped) pointwise convolution. All streams are combined through a stable, fully differentiable dynamic convolution—no virtual tokens, no auxiliary heads, no Top-$K$ or low-rank—yielding parameter-efficient pipeline-level capacity scaling with hardware-friendly convolutional primitives.

Our contributions are as follows:

- **Theoretical:** We establish the formal equivalence between MoE-style mixture layers and dynamic convolutions with variable kernels (Theorem 4.1). This provides a unified theoretical framework for analyzing and designing sparse, conditional computation models, beyond FFN-level MoE.

- **Methodological:** We introduce **cMoLLM**, a pipeline-level convolutionally-gated mixture whose kernel-sharing and stream structure follow directly from the convolution view. It achieves parameter-efficient scaling and training stability without virtual tokens, auxiliary prediction branches, Top-$K$ routing, or low-rank factorization.

- **Empirical:** On GPT-2–style models trained on FineWeb, cMoLLM matches or improves perplexity, GLUE, and SQuAD over dense baselines under comparable cost, with better stream utilization, more stable training dynamics, and favorable scaling (see Section 5.3) compared to ParaScale- and AltUp-style pipeline mixtures.

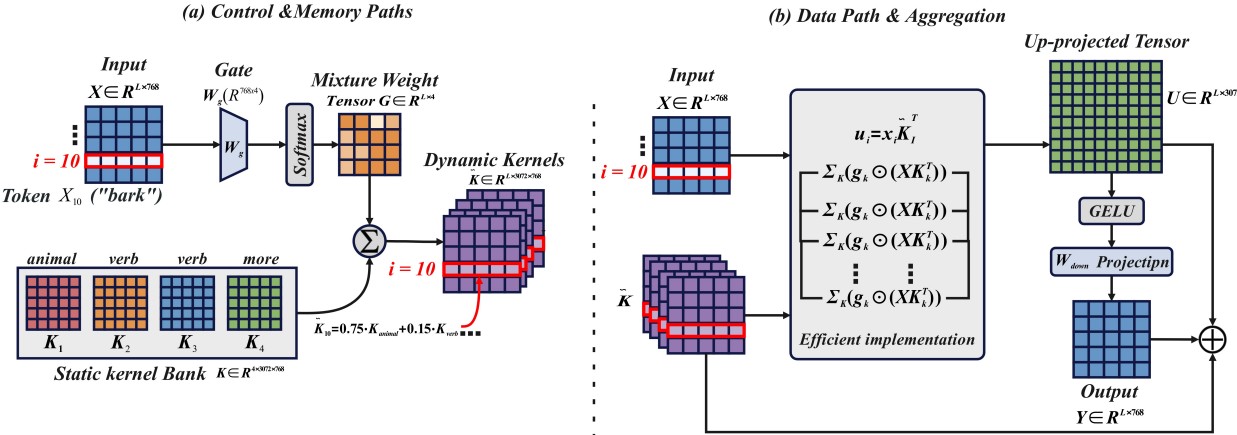

*Figure 2.* **Architecture (Fig. 2).** cMoLLM block: input $\mathbf{X}$ passes through a gating network to produce soft mixture weights $\{g_k\}$; each stream has a $1 \times 1$ kernel $\mathbf{K}_k$; the effective kernel $\widetilde{\mathbf{K}}(\mathbf{x}) = \sum_k g_k(\mathbf{x})\mathbf{K}_k$ is applied via grouped $1 \times 1$ convolution. No virtual tokens, no Top-$K$, no auxiliary heads.

## 2. Related Work

**Mixture-of-Experts and Pipeline-Level Scaling.** Classical MoE approaches (Shazeer et al., 2017; Fedus et al., 2022; Lepikhin et al., 2021) sparsify the **feed-forward network (FFN)** layer using Top-$K$ routing, where each token is routed to a sparse subset of expert FFN blocks. Large-scale MoE systems such as GLaM (Du et al., 2022), BASE Layers (Lewis et al., 2021), DeepSpeed-MoE (Rajbhandari et al., 2022), and Vision MoE models (Riquelme et al., 2021) focus primarily on FFN-level sparsity and system-level optimizations for training and inference. In contrast, we target pipeline-level mixture over entire LLM streams, leveraging the MoE–dynamic convolution equivalence (Section 4.1) to design convolutionally-gated routing without Top-$K$ truncation. The most directly relevant prior work includes **ParaScale** (Chen et al., 2025), which scales capacity via parallel virtual token streams but increases compute and suffers from gradient collapse, and **AltUp** (Baykal et al., 2023), which uses a hand-designed auxiliary prediction branch but converges slowly. Table 1 provides a systematic comparison; cMoLLM avoids virtual tokens and auxiliary heads, using fully differentiable dynamic convolution for stable, parameter-efficient scaling.

**Dynamic Convolution and Conditional Computation.** Dynamic convolution (Jia et al., 2016; Yang et al., 2019; Chen et al., 2020; Wu et al., 2019; Ma et al., 2020; Zhang et al., 2020) adapts convolution kernels based on input, typically within CNN backbones or sequence models. More broadly, conditional computation and routing networks (Bengio et al., 2013; Rosenbaum et al., 2018; McGill & Perona, 2017) learn to select computation paths depending on inputs. Our key theoretical contribution is establishing the formal equivalence between MoE-style mixture layers and dynamic convolutions with variable kernels

(Theorem 4.1), which enables us to design a pipeline-level, convolutionally-gated mixture over LLM streams with explicit probabilistic routing and load-balancing objectives.

**Parameter-Efficient Methods and Surveys.** Parameter-efficient fine-tuning methods such as adapters (Houlsby et al., 2019) and LoRA (Hu et al., 2022) add or reparameterize a small set of trainable weights on top of frozen backbones, primarily targeting efficient fine-tuning. cMoLLM instead modifies the pretraining architecture to improve the capacity–compute trade-off; these approaches are complementary and could in principle be combined. Comprehensive surveys of MoE models and their applications in LLMs and beyond are provided by Cai et al. (2025); Mu & Lin (2025); Liu et al. (2026), which situate our contribution within the broader MoE landscape.

## 3. Preliminaries

We first introduce the notation used throughout the paper and briefly review the core concepts that underpin our formulation.

### 3.1. Notation and Assumptions

We use bold lowercase letters (e.g., $\mathbf{x}$) for vectors, bold uppercase letters (e.g., $\mathbf{W}$) for matrices, and calligraphic letters (e.g., $\mathcal{S}$) for sets. The hidden dimension of the Transformer is $d$, the intermediate FFN dimension is $d_{\text{ff}}$, the sequence length is $L$ and the number of experts/streams is $N$. For $\mathbf{X} \in \mathbb{R}^{L \times d}$, the $i$-th token is denoted by $\mathbf{x}_i$.

Throughout, we make the following mild assumptions.

**Assumption 3.1** (Bounded Inputs). There exists $R > 0$ such that $\|\mathbf{x}\|_2 \leq R$ for all token representations $\mathbf{x}$ encountered during training and evaluation.

**Assumption 3.2** (Lipschitz Nonlinearity). *The activation function $\sigma$ is $L_\sigma$-Lipschitz and satisfies $\sigma(0) = 0$ (e.g., ReLU or GELU).*

These conditions are standard in theoretical analyses of deep networks and MoE architectures, and they suffice for establishing the complexity and stability results in this paper.

### 3.2. Transformer Feed-Forward Network

A standard Transformer FFN applies two linear projections with a nonlinearity (Vaswani et al., 2017):

$$\text{FFN}(\mathbf{x}) = \mathbf{W}_2\, \sigma(\mathbf{W}_1\mathbf{x} + \mathbf{b}_1) + \mathbf{b}_2, \qquad (1)$$

where $\mathbf{x} \in \mathbb{R}^d$ is the input token representation, $\mathbf{W}_1 \in \mathbb{R}^{d_{\text{ff}} \times d}$, $\mathbf{W}_2 \in \mathbb{R}^{d \times d_{\text{ff}}}$, and $\sigma$ is typically GELU or ReLU. The FFN is applied independently to each token.

### 3.3. Mixture-of-Experts Layer

An MoE layer replaces a single feed-forward network with a collection of $N$ experts $\{E_k\}_{k=1}^N$ and a gating function $G$ that routes each input to a subset of experts (Shazeer et al., 2017):

$$\text{MoE}(\mathbf{x}) = \sum_{k=1}^N g_k(\mathbf{x})\, E_k(\mathbf{x}), \qquad (2)$$

where $g(\mathbf{x}) \in \mathbb{R}^N$ denotes the routing weights produced by the gate. In practice, only the top-$K$ entries of $g(\mathbf{x})$ are nonzero, yielding sparse computation. Each expert $E_k$ is typically a two-layer FFN with its own parameters.

### 3.4. Pointwise Convolution as a Linear Transform

A pointwise ($1\times1$) convolution applies the same linear map across token positions. Given an input sequence $\mathbf{X} \in \mathbb{R}^{L \times d_{\text{in}}}$ and a kernel $\mathbf{K} \in \mathbb{R}^{d_{\text{out}} \times d_{\text{in}}}$, the operation is given by

$$\text{Conv}_{1\times1}(\mathbf{X}; \mathbf{K}) = \mathbf{X}\mathbf{K}^\top, \qquad (3)$$

which corresponds to applying an identical token-wise linear projection from $\mathbb{R}^{d_{\text{in}}}$ to $\mathbb{R}^{d_{\text{out}}}$. This observation serves as a key building block for our subsequent analysis.

## 4. Method: cMoLLM

We now present our main theoretical result and the cMoLLM architecture.

### 4.1. MoE as Dynamic Convolution: A Formal Equivalence

We show that MoE layers admit an equivalent interpretation as dynamic pointwise convolutions. In this subsection we focus on linear experts and the structure of the router; extensions to nonlinear experts are discussed in Section A and summarized in Theorem 4.2.

**Theorem 4.1** (MoE–Dynamic Convolution Equivalence). *Let $\mathbf{x} \in \mathbb{R}^d$ denote a token representation. Consider an MoE layer with $N$ linear experts $E_k(\mathbf{x}) = \mathbf{x}\mathbf{W}_k^\top$, where $\mathbf{W}_k \in \mathbb{R}^{d_{\text{out}} \times d}$, and routing weights $\{g_k(\mathbf{x})\}_{k=1}^N$ satisfying $\sum_{k=1}^N g_k(\mathbf{x}) = 1$ for all $\mathbf{x}$. Then the MoE output can be written as a dynamic $1\times1$ convolution:*

$$\text{MoE}(\mathbf{x}) = \text{Conv}_{1\times1}\Big(\mathbf{x}; \tilde{\mathbf{K}}(\mathbf{x})\Big) = \mathbf{x}\,\tilde{\mathbf{K}}(\mathbf{x})^\top, \quad (4)$$

*where the effective kernel is given by*

$$\tilde{\mathbf{K}}(\mathbf{x}) := \sum_{k=1}^N g_k(\mathbf{x})\, \mathbf{W}_k. \qquad (5)$$

*Since $\tilde{\mathbf{K}}(\mathbf{x})$ depends on the input $\mathbf{x}$, the MoE layer is precisely a form of dynamic convolution.*

*Proof.* By linearity of matrix multiplication:

$$\sum_{k=1}^N g_k(\mathbf{x}) \cdot (\mathbf{x}\mathbf{W}_k^\top) = \mathbf{x}\Big(\textstyle\sum_{k=1}^N g_k(\mathbf{x})\mathbf{W}_k\Big)^\top$$
$$= \text{Conv}_{1\times1}\Big(\mathbf{x}; \textstyle\sum_{k=1}^N g_k(\mathbf{x})\mathbf{W}_k\Big). \qquad (6)$$

The weighted sum of expert weight matrices forms a dynamic kernel that varies with $\mathbf{x}$. (Classical MoE uses Top-$K$ routing so only $K$ terms are nonzero; we use soft mixing over all streams.) See Section A for extension to nonlinear experts.

**Corollary 4.2** (Nonlinear Two-Layer Experts). *Under Theorem 3.2, consider two-layer experts of the form $E_k(\mathbf{x}) = \sigma(\mathbf{x}\mathbf{W}_{k,1}^\top)\mathbf{W}_{k,2}^\top$ with routing weights $\{g_k(\mathbf{x})\}_{k=1}^N$ satisfying $\sum_k g_k(\mathbf{x}) = 1$. Then the MoE layer can still be written as a dynamic $1\times1$ convolution with an input-dependent effective kernel that incorporates a data-dependent mask induced by $\sigma$; see Section A for details.*

Assumptions 3.1 and 3.2 are not needed for Theorem 4.1 itself, but they will be used in our later complexity, stability, and toy-model analyses that build on this equivalence.

*Remark* 4.3 (Key Differences from Standard Convolution). While Theorem 4.1 reveals structural similarity, standard convolutional layers differ from explicit MoE in three aspects:

1. **Static vs. Learnable Gating.** In CNN, activation functions (e.g., ReLU) act as fixed, non-learnable gates determined by the sign of pre-activations. In MoE, the router is a trainable subnetwork whose routing policy is learned end-to-end.

2. **Unnormalized vs. Normalized Weights.** CNN activations are unnormalized; MoE gating weights satisfy $\sum_k g_k = 1$, representing a probability distribution over experts.

3. **Dense vs. Sparse Computation.** Standard CNN layers apply all filters densely; classical MoE routes each token to only the top-$K$ experts. We use soft mixture weights over all streams (no Top-$K$), so computation scales with $N$ streams but remains a single dynamic convolution.

This equivalence motivates our approach: we implement pipeline-level mixture as an explicit dynamic convolution, leveraging efficient convolutional primitives.

### 4.2. cMoLLM Block Design

Based on Theorem 4.1, we design **cMoLLM** as a pipeline-level mixture. Figure 2 illustrates the architecture: gating, per-stream kernels, and dynamic convolution.

**Gating Network.** Given input $\mathbf{X} \in \mathbb{R}^{L \times d}$, a lightweight MLP produces routing logits:

$$\begin{aligned}
\mathbf{H}_{\text{flat}} &= \text{Flatten}(\mathbf{Y}_{\text{streams}}) \in \mathbb{R}^{L \times (N \cdot d)}, \\
\mathbf{H}_{\text{mid}} &= \sigma\big(\mathbf{H}_{\text{flat}}\mathbf{W}_g^{(1)} + \mathbf{b}_g^{(1)}\big) \in \mathbb{R}^{L \times d}, \\
\mathbf{H}_{\text{logits}} &= \mathbf{H}_{\text{mid}}\mathbf{W}_g^{(2)} + \mathbf{b}_g^{(2)} \in \mathbb{R}^{L \times N},
\end{aligned} \quad (7)$$

where $\mathbf{W}_g \in \mathbb{R}^{d \times N}$. We apply a softmax over streams to obtain mixture weights:

$$g_k(\mathbf{x}) = \text{softmax}(\mathbf{h})_k, \quad k \in [N]. \quad (8)$$

This fully differentiable gating avoids discrete Top-$K$ truncation; all streams participate via soft weights. We support multiple gating variants (simple, context_aware, multi_head, adaptive), evaluated in Section 5.

**Expert Kernel Generator.** Each stream $k$ has its own $1 \times 1$ convolutional kernel $\mathbf{K}_k \in \mathbb{R}^{d_{\text{ff}} \times d}$. The effective kernel is their mixture weighted by $g_k(\mathbf{x})$; we use no low-rank factorization or shared base in our main setup.

**Dynamic Grouped Convolution.** Let a set of mixture weights $\{g_k(\mathbf{x})\}_{k=1}^N$ be given. We form a dynamic mixture of convolutional kernels by

$$\tilde{\mathbf{K}}(\mathbf{x}) = \sum_{k=1}^{N} g_k(\mathbf{x})\,\mathbf{K}_k. \quad (9)$$

Practically, we compute branch-wise outputs in parallel and fuse them:

$$\mathbf{X}_k = \text{Conv}_k(\mathbf{X}; \mathbf{K}_k), \quad k \in [N], \quad (10)$$

and concatenate the per-branch outputs along the representation dimension:

$$\mathbf{X}_{\text{input}} = [\mathbf{X}_1, \mathbf{X}_2, \dots, \mathbf{X}_N] \in \mathbb{R}^{L \times N \times d}, \quad (11)$$

where $L$ is the sequence length and $d$ is the feature dimension (hidden size).

To model interactions among the branches, we apply a transformer block to the stream of parallel outputs:

$$\mathbf{Y}_{\text{stream}} = \text{TRANSFORMERS}\big(\mathbf{X}_{\text{input}}; \tilde{\mathbf{K}}(\mathbf{X})\big). \quad (12)$$

Here $\text{TRANSFORMERS}(\cdot)$ denotes one or more Transformer layers, and the attention (or feed-forward) computations may be conditioned on the dynamic kernels $\tilde{\mathbf{K}}(\mathbf{X})$.

**ParaScale Output Aggregation.** Let the hidden states have shape $\mathbf{H} \in \mathbb{R}^{S \times B \times H}$. If the ParaScale module uses $N_{\text{ps}} > 1$ parallel scales, we first reorganize the hidden states along the parallel dimension:

$$\mathbf{H} \to \hat{\mathbf{H}} = \text{reshape}\big(\mathbf{H},\, S,\, B/N_{\text{ps}},\, N_{\text{ps}},\, H\big), \quad (13)$$

and flatten the parallel axis:

$$\mathbf{H}_{\text{flat}} = \text{reshape}\big(\hat{\mathbf{H}},\, S,\, B/N_{\text{ps}},\, N_{\text{ps}}\,H\big). \quad (14)$$

We compute an aggregation weight via a learned layer:

$$\boldsymbol{\alpha}_{s,b} = \text{softmax}\big(\mathbf{W}_{\text{agg}}\,\mathbf{H}_{\text{flat}}^{s,b,*}\big) \in \mathbb{R}^{N_{\text{ps}}}, \quad \|\boldsymbol{\alpha}_{s,b}\|_1 = 1. \quad (15)$$

**Smoothness and Gradient-Aware Measures.** 1) Per-step attention smoothing. Let $\boldsymbol{\alpha}_{s,b}$ be the per-step, per-sample attention over the $N_{\text{ps}}$ parallel scales. We define the smoothed attention as

$$\boldsymbol{\alpha}_{s,b}^{\text{smooth}} = (1-\beta)\,\boldsymbol{\alpha}_{s,b} + \beta\,\frac{1}{N_{\text{ps}}}\,\mathbf{1}, \qquad \beta \in [0,1]. \quad (16)$$

2) Gradient-aware aggregation. The weighted sum uses the smoothed weights:

$$\mathbf{H}_{\text{weighted}}^{s,b} = \sum_{i=1}^{N_{\text{ps}}} \alpha_{s,b}^{(i)\,\text{smooth}}\,\mathbf{H}_{\text{flat}}^{s,b,i}, \quad (17)$$

followed by a linear projection and a mean residual term:

$$\mathbf{H}_{\text{agg}}^{s,b} = \mathbf{F}\,\mathbf{H}_{\text{weighted}}^{s,b} + \mathbf{H}_{\text{mean}}^{s,b}, \quad \mathbf{H}_{\text{out}}^{s,b} = \text{Proj}\big(\mathbf{H}_{\text{agg}}^{s,b}\big). \quad (18)$$

### 4.3. cMoLLM Forward Pass: Pseudocode

Algorithm 1 gives pseudocode for a single Transformer layer with a cMoLLM block (no Top-$K$, no low-rank).

### 4.4. CNN as a Constrained Implicit MoE

The equivalence in Theorem 4.1 suggests that standard convolutional layers can be interpreted as special cases of MoE under strong constraints. We formalize this intuition for a single linear or convolutional layer.

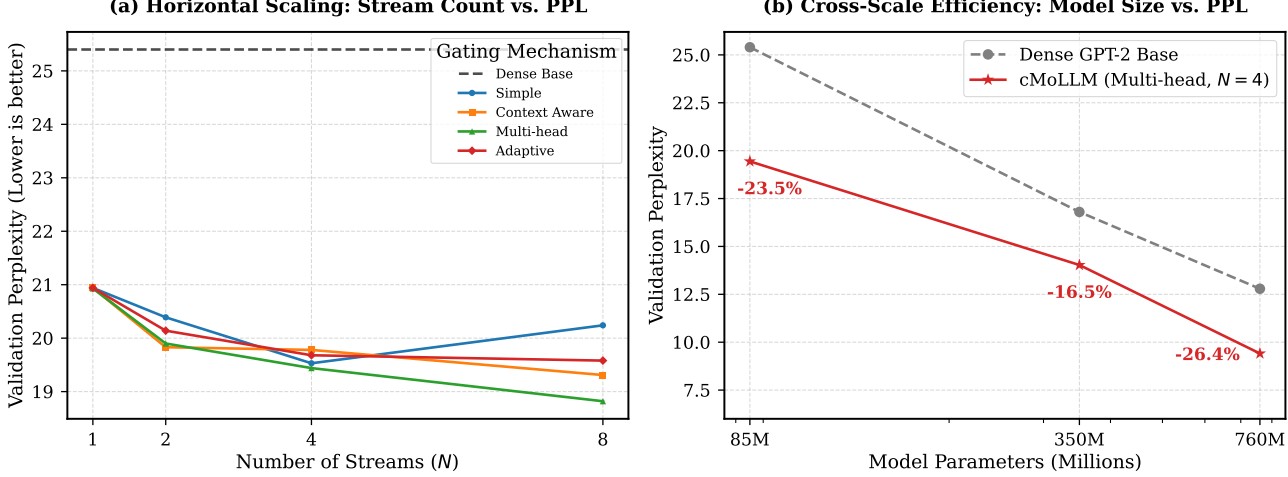

*Figure 3.* **Experimental results (Fig. 3).** Left: per-stream gating weight distribution across layers. Right: validation loss and perplexity vs. stream count $n$ or training steps. cMoLLM maintains stable utilization and scaling gains.

---

**Algorithm 1** Forward pass of a Transformer layer with cMoLLM

---

1: **Input:** $\mathbf{X} \in \mathbb{R}^{L \times d}$, number of streams $N$, kernels $\{\mathbf{K}_k\}_{k=1}^N$, $\mathbf{W}_{\text{down}}$
2: **Self-Attention:** $\mathbf{H} \leftarrow \text{SelfAttention}(\mathbf{X})$
3: **Gating logits:** $\mathbf{G} \leftarrow \mathbf{H}\mathbf{W}_g + \mathbf{b}_g \in \mathbb{R}^{L \times N}$
4: **Soft mixture weights:** $g_{i,k} \leftarrow \text{softmax}(\mathbf{G}_{i,:})_k$ for each token $i$, stream $k$
5: **Mixed kernel:** $\widetilde{\mathbf{K}}_i \leftarrow \sum_{k=1}^N g_{i,k}\,\mathbf{K}_k$
6: **Dynamic** $1 \times 1$ **conv:** $\mathbf{U}_i \leftarrow \mathbf{H}_i\,\widetilde{\mathbf{K}}_i^\top$
7: **Nonlinearity & down-proj:** $\mathbf{Y}_i \leftarrow \mathbf{W}_{\text{down}}\,\sigma(\mathbf{U}_i)$
8: **Output:** $\mathbf{Y}$ (residual + norm as usual)

---

**Proposition 4.4** (CNN as Implicit Constrained MoE). *Consider a linear layer* $\mathbf{y} = \mathbf{W}\mathbf{x}$ *with* $\mathbf{W} \in \mathbb{R}^{d_{\text{out}} \times d}$ *and activation* $\sigma$ *satisfying Theorem 3.2. Let* $\mathbf{w}_j^\top$ *denote the $j$-th row of* $\mathbf{W}$ *and define per-output "experts"* $E_j(\mathbf{x}) = \sigma(\mathbf{w}_j^\top \mathbf{x})$ *for* $j \in [d_{\text{out}}]$. *Then the layer can be written as an MoE of* $d_{\text{out}}$ *experts with:*

1. *static, parameter-free gating determined solely by the sign pattern of pre-activations;*

2. *unnormalized gating weights given by* $\sigma(\mathbf{w}_j^\top \mathbf{x})$;

3. *dense computation, as all experts are evaluated for every input.*

*In particular, a standard CNN layer implements an implicit, highly constrained MoE over its output channels.*

*Proof.* Write the layer output as $\mathbf{y} = \sigma(\mathbf{W}\mathbf{x})$, where $\sigma$ is applied element-wise. For each output dimension $j$, we have $y_j = \sigma(\mathbf{w}_j^\top \mathbf{x})$, which we interpret as the output of

expert $E_j$. Define the gating weight $g_j(\mathbf{x}) = 1$ for all $j$ and inputs. Then the overall output can be written as

$$\mathbf{y} = \sum_{j=1}^{d_{\text{out}}} g_j(\mathbf{x})\,E_j(\mathbf{x}) = \sum_{j=1}^{d_{\text{out}}} E_j(\mathbf{x}), \tag{19}$$

which matches the standard layer. The nonlinearity $\sigma$ induces a data-dependent binary mask $\mathbb{I}(\mathbf{w}_j^\top \mathbf{x} > 0)$ on each expert, yielding a form of fixed, unnormalized gating as discussed in Theorem 4.3. Unlike explicit MoE, all experts are always evaluated, so computation is dense.

This proposition connects classical CNNs to MoE: they sit at one end of a spectrum with static, dense, and unnormalized routing, whereas cMoLLM uses learned, normalized routing via dynamic convolutions (soft mixture over all streams, no Top-$K$). Viewed through this lens, standard CNNs are implicit, constrained MoE models, and **cMoLLM** can be seen as relaxing these constraints—moving along a continuum from fixed, dense routing to learned, normalized, and capacity-controllable routing over pipeline-level streams.

### 4.5. Gating Variants (Convolution Types)

We implement four gating variants corresponding to the `simple`, `context_aware`, `multi_head`, and `adaptive` options evaluated in Section 5.

#### 4.5.1. SIMPLE GATED CONVOLUTION

Two parallel convolutions are applied: one for feature extraction and one for gate generation. The feature output is element-wise multiplied by the sigmoid-activated gate output. Gating is local and token-wise, with no global context.

### 4.5.2. CONTEXT-AWARE GATED CONVOLUTION

A context analyzer (global average pooling plus an MLP with 4:1 compression) produces channel-wise modulation weights from the full sequence; these are combined with the local gate convolution output. Gating thus depends on both local patterns and sequence-level statistics.

### 4.5.3. MULTI-HEAD GATED CONVOLUTION

Multiple independent gate convolutions operate on the same input; each head produces gates for a disjoint subset of output channels. Outputs are concatenated and fused by a learned layer. Heads can specialize in different temporal scales or aspects of the input.

### 4.5.4. ADAPTIVE GATED CONVOLUTION

An adaptive network (global pooling plus a two-layer MLP) outputs three bounded parameters (sensitivity, bias, magnitude) from global input statistics. The gate is sigmoid(scaled gate-conv output + bias) times magnitude. Gating behavior is adjusted per sequence (e.g., dynamic range or complexity).

### 4.6. Training Strategies

**Load Balancing Loss.** To encourage balanced use of streams, we add an auxiliary loss (Fedus et al., 2022):

$$\mathcal{L}_{\text{bal}} = \alpha \cdot N \cdot \sum_{k=1}^{N} p_k^2, \tag{20}$$

where $p_k = \frac{1}{L} \sum_{i=1}^{L} g_k(\mathbf{x}_i)$ is the average gating probability for stream $k$ over the sequence. Minimizing $\sum_k p_k^2$ discourages collapse onto a few streams. We use $\alpha = 0.01$ in experiments.

**Total Loss.** The total training objective is $\mathcal{L} = \mathcal{L}_{\text{LM}} + \mathcal{L}_{\text{bal}}$; we use no kernel regularization, progressive sparsification, or low-rank terms.

### 4.7. Computational Complexity Analysis

We compare per-token cost of dense FFN, Top-$K$ MoE, and cMoLLM. Let $\text{FLOPs}(\cdot)$ denote leading-order FLOPs per token.

Dense FFN: $\text{FLOPs}_{\text{dense}} \approx 2dd_{\text{ff}}$ (up- and down-projections). Top-$K$ MoE with $N$ experts: $\text{FLOPs}_{\text{moe}} \approx 2Kdd_{\text{ff}} + \text{FLOPs}_{\text{gate}}$.

In cMoLLM we compute $\mathbf{U}_i = \sum_{k=1}^{N} g_{i,k}(\mathbf{H}_i \mathbf{K}_k^\top)$ then $\mathbf{W}_{\text{down}} \sigma(\mathbf{U}_i)$. The $N$ stream products cost $N \cdot d \cdot d_{\text{ff}}$, and the down-projection $d_{\text{ff}} \cdot d$. Thus

$$\text{FLOPs}_{\text{cMoLLM}} \approx (N+1)dd_{\text{ff}} + \text{FLOPs}_{\text{gate}}. \tag{21}$$

For small $N$ (e.g., $N=4$), this is on the same order as dense, while affording $N$ distinct kernels and pipeline-level mixture. Convolution primitives often yield favorable throughput in practice.

Combining the above, we can summarize the horizontal scaling behavior as follows. Fix a compute budget $\text{FLOPs}_0$ and FFN dimension $d_{\text{ff}}$. Then there exists a constant $C$ (absorbing gating overhead) such that

$$\text{FLOPs}_{\text{cMoLLM}} \leq C \cdot \text{FLOPs}_{\text{dense}} \quad \text{whenever } N \leq C-1, \tag{22}$$

while the number of distinct kernels (and thus the effective capacity of the mixture) grows linearly with $N$. Top-$K$ MoE trades compute for sparsity: increasing $N$ at fixed $K$ primarily increases parameter count but leaves per-token compute approximately $O(Kdd_{\text{ff}})$. cMoLLM therefore realizes a horizontal scaling law: for bounded $N$, we scale capacity roughly linearly in $N$ while keeping per-token compute within a constant factor of the dense baseline.

### 4.8. Intuitive Summary

We summarize cMoLLM intuitively. MoE-style mixture linearly combines expert outputs; Theorem 4.1 shows this is equivalent to mixing $1\times1$ convolution kernels—i.e., dynamic convolution with variable kernels. We approximate this with per-stream kernels $\mathbf{K}_k$ and soft mixture weights $g_k(\mathbf{x})$; no Top-$K$, no low-rank.

cMoLLM is implemented with standard $1\times1$ convolutions and a lightweight gating network. Recipe: (i) keep the Transformer backbone; (ii) replace FFN blocks with cMoLLM blocks (streams + gating); (iii) tune $N$ to trade capacity vs. compute. Experiments (Section 5) show better perplexity and GLUE at similar compute, with more stable stream utilization than ParaScale- and AltUp-style pipeline mixtures. In addition, a cluster-structured toy model in Section B formalizes when routing (and by equivalence, dynamic convolution) can achieve Bayes-optimal performance while any single linear classifier suffers a nontrivial error floor, illustrating the potential benefits of cMoLLM-style conditional computation.

## 5. Experiments

We evaluate cMoLLM on language modeling, comparing to a dense GPT-2 baseline under matched training setups.

### 5.1. Experimental Setup

**Model Configuration.** We adopt a GPT-2-style architecture (Radford et al., 2019) as our base model. We evaluate at three scales (Table 4): small ( 85M), medium ( 350M), and large ( 760M) parameters, varying layers, hidden size, and attention heads. Table 5 gives shared hyperparameters.

*Table 2.* Full experimental results (3 seeds; mean ± std): validation loss, perplexity (PPL), GLUE (%), SQuAD v2 (%). "conv" = gating variant; $n$ = streams. **Bold** = best (SOTA) in column.

| conv | $n$ | Loss ↓ | PPL ↓ | GLUE (%) ↑ | SQuAD v2 (%) ↑ |
|---|---|---|---|---|---|
| base | – | 3.23±0.03 | 25.40±0.65 | 42.45±0.99 | 50.07±0.65 |
| simple | 1 | 3.04±0.03 | 20.94±0.22 | 44.16±0.56 | 51.13±0.54 |
| simple | 2 | 3.01±0.02 | 20.39±0.56 | 44.62±1.02 | 52.08±0.95 |
| simple | 4 | 2.97±0.04 | 19.53±0.49 | 44.92±0.87 | 55.71±1.30 |
| simple | 8 | 3.00±0.03 | 20.24±0.12 | 44.68±0.33 | 55.95±0.51 |
| context_aware | 1 | 3.04±0.03 | 20.94±0.41 | 44.16±0.42 | 51.12±1.28 |
| context_aware | 2 | 2.98±0.03 | 19.83±0.55 | 46.79±1.21 | 52.13±1.13 |
| context_aware | 4 | 2.98±0.02 | 19.78±0.12 | 47.12±0.25 | 55.64±0.87 |
| context_aware | 8 | 2.96±0.01 | 19.31±0.68 | **48.98±0.64** | 56.11±0.73 |
| multi_head | 1 | 3.04±0.01 | 20.93±0.22 | 44.16±1.18 | 51.18±0.66 |
| multi_head | 2 | 2.99±0.05 | 19.90±0.28 | 47.78±0.45 | 52.98±1.35 |
| multi_head | 4 | 2.96±0.04 | 19.44±0.41 | 48.02±0.79 | 56.37±1.56 |
| multi_head | 8 | **2.93±0.02** | **18.82±0.37** | 48.82±1.01 | **57.06±0.52** |
| adaptive | 1 | 3.04±0.05 | 20.94±0.46 | 44.16±0.98 | 51.28±0.88 |
| adaptive | 2 | 3.00±0.02 | 20.14±0.26 | 46.64±0.43 | 52.43±1.45 |
| adaptive | 4 | 2.97±0.03 | 19.68±0.71 | 47.03±0.36 | 55.89±1.39 |
| adaptive | 8 | 2.97±0.03 | 19.58±0.65 | 47.81±0.75 | 56.74±0.72 |

*Table 3.* Scaling experiments (3 seeds; mean ± std) across model sizes (Table 4). mh = multi-head cMoLLM; base = dense baseline. **Bold** = SOTA in column.

| type | Loss ↓ | PPL ↓ | GLUE ↑ | SQuAD v2 ↑ |
|---|---|---|---|---|
| base-85M | 3.23±0.03 | 25.40±0.65 | 42.45±0.99 | 50.07±0.65 |
| base-350M | 2.82±0.04 | 16.80±0.47 | 50.68±1.08 | 59.24±1.24 |
| base-760M | 2.55±0.05 | 12.79±0.28 | 55.74±0.46 | 62.91±0.74 |
| mh-85M | 2.96±0.03 | 19.44±0.56 | 48.02±0.84 | 56.37±0.54 |
| mh-350M | 2.64±0.03 | 14.03±0.37 | 53.82±1.03 | 61.08±1.26 |
| mh-760M | **2.24±0.02** | **9.41±0.45** | **58.45±0.77** | **64.56±1.32** |

*Table 4.* Model scales used in scaling experiments. Config 1 = small (85M), Config 2 = medium (350M), Config 3 = large (760M).

| Config | Layers | Hidden | Heads | Params |
|---|---|---|---|---|
| 1 | 12 | 768 | 12 | ∼85M |
| 2 | 24 | 1024 | 16 | ∼350M |
| 3 | 24 | 1536 | 24 | ∼760M |

*Table 5.* Shared training hyperparameters across all variants.

| Hyperparameter | Value |
|---|---|
| Base architecture | GPT-2 |
| FFN dimension ($d_{\text{ff}}$) | 3072 |
| Sequence length | 4096 |
| Learning rate | $6 \times 10^{-5}$ |

**Training Data.** We train on FineWeb (Penedo et al., 2024), a large-scale curated web corpus.

**Baselines.** Our main baseline is the dense GPT-2 model in which all layers are standard Transformer blocks. For cMoLLM we keep the backbone identical and replace the FFN stack with convolutionally-gated mixture streams.

**Scaling Factors.** We use scaling factors $n \in \{1, 2, 4, 8\}$, where $n$ is the number of parallel cMoLLM streams.

**Gating Variants.** We evaluate gating types: simple, context_aware, multi_head, and adaptive.

**Reporting.** All experiments use **3 random seeds**. We report **mean ± standard deviation (std)** for all metrics in tables.

## 5.2. Main Results

We report language modeling (loss, perplexity), GLUE, and SQuAD v2 accuracy for each cMoLLM variant (Table 2).

**Experiment figures.** Figure 3 shows stream utilization and horizontal scaling: per-stream gating weight distribution across layers (left) and validation loss / perplexity vs. stream

count $n$ or training steps (right). Together with Tables 2 and 3, the figure confirms balanced stream usage and gains as $n$ increases over an optimal range, without collapse.

### 5.3. Scaling Laws

We consolidate scaling behavior along two axes: (i) *horizontal* scaling in the number of streams $N$ at fixed model size (as in Section 4.7), and (ii) *model-size* scaling from 85M to 760M parameters.

**Horizontal scaling (stream count).** As shown in Section 4.7, cMoLLM satisfies a horizontal scaling law: for bounded $N$, effective capacity grows roughly linearly in $N$ while per-token FLOPs remain within a constant factor of the dense baseline (Equation (21)). Figure 3 and Table 2 confirm that validation loss and perplexity improve as $n$ increases from 1 to 4–8, with best loss/PPL at `multi_head` $n=8$; beyond that, some gating variants show mild over-streaming (e.g., `simple` at $n=8$), while `multi_head` and `adaptive` remain stable.

**Model-size scaling.** Table 3 reports results across the three scales in Table 4 (85M, 350M, 760M). At every scale, multi-head cMoLLM (`mh`) outperforms the dense baseline (`base`) on loss, PPL, GLUE, and SQuAD v2. The gains are consistent with standard scaling: loss and perplexity decrease as model size increases, and the relative advantage of cMoLLM over the dense baseline is preserved (e.g., `mh-760M` achieves SOTA across all four metrics). This supports that the convolutionally-gated pipeline mixture scales favorably with both stream count and parameter count under matched training setups.

## 6. Conclusion

We have presented **cMoLLM**, a convolutionally-gated mixture-of-LLMs that scales capacity at the pipeline level. Our main theoretical contribution is the formal equivalence between MoE-style mixture layers and dynamic convolutions with variable kernels, providing a unified framework for analyzing and designing sparse, conditional computation beyond FFN-level MoE. cMoLLM instantiates this via per-stream kernels and soft gating—no Top-$K$, no low-rank, no virtual tokens or auxiliary heads—yielding parameter-efficient, stable pipeline-level scaling.

Experiments on GPT-2–style models trained on FineWeb show that cMoLLM improves perplexity, GLUE, and SQuAD under matched compute, with better stream utilization and training stability than ParaScale- and AltUp-style pipeline mixtures. Scaling laws (horizontal scaling in stream count and model-size scaling) are analyzed in Section 5.3. The convolution-based design enables efficient implementation and favorable scaling in practice. Extended results on downstream tasks and scaling are provided in Section B.3.

**Limitations and Future Work.** Experiments are at GPT-2 scale; validation at 7B+ parameters is needed. Future work: (i) scale cMoLLM to larger models and distributed training; (ii) extend the MoE–convolution equivalence to attention; (iii) combine with retrieval or other conditional compute mechanisms.

## Impact Statement

This work introduces cMoLLM, a convolutionally-gated mixture-of-LLMs that scales capacity efficiently without auxiliary routing mechanisms. **Positive impacts:** Parameter-efficient scaling reduces training/inference costs and energy consumption; the convolution-based design is hardware-friendly. **Risks:** Misuse concerns (e.g., misleading content) remain, though we introduce no new failure modes beyond existing LLMs. Experiments are limited to GPT-2 scale ( 760M parameters); validation at 7B+ is needed. **Summary:** We believe the net effect is beneficial by improving LLM scaling efficiency.

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

*

## A. Proof of Theorem 4.1: Extension to Nonlinear Experts

We extend the equivalence result to two-layer experts with nonlinear activations.

**Setup.** Let $E_k(\mathbf{x}) = \sigma(\mathbf{x}\mathbf{W}_{k,1}^\top)\mathbf{W}_{k,2}^\top$ be a two-layer expert with activation $\sigma$ (e.g., ReLU, GELU). The MoE output is:

$$\text{MoE}(\mathbf{x}) = \sum_{k=1}^{N} g_k(\mathbf{x})\, \sigma(\mathbf{x}\mathbf{W}_{k,1}^\top)\mathbf{W}_{k,2}^\top. \tag{23}$$

**Linear Case.** When $\sigma = \text{id}$ (identity), we recover the result in Theorem 4.1:

$$\sum_k g_k(\mathbf{x})\, \mathbf{x}\mathbf{W}_{k,1}^\top \mathbf{W}_{k,2}^\top = \mathbf{x}\Big(\textstyle\sum_k g_k(\mathbf{x})\, \mathbf{W}_{k,2}\mathbf{W}_{k,1}\Big)^\top$$

$$= \text{Conv}_{1\times1}\Big(\mathbf{x};\, \textstyle\sum_k g_k(\mathbf{x})\, \mathbf{W}_{k,2}\mathbf{W}_{k,1}\Big). \tag{24}$$

**Nonlinear Case (ReLU).** For $\sigma = \text{ReLU}$, we can rewrite element-wise:

$$\text{ReLU}(z) = \mathbb{I}(z > 0) \cdot z, \tag{25}$$

where $\mathbb{I}(\cdot)$ is the indicator function. The ReLU acts as a data-dependent binary gate.

Define the diagonal masking matrix:

$$\mathbf{M}_k(\mathbf{x}) = \text{diag}\big(\mathbb{I}(\mathbf{x}\mathbf{W}_{k,1}^\top > 0)\big) \in \{0,1\}^{d_{\text{ff}} \times d_{\text{ff}}}. \tag{26}$$

Then:

$$\sigma(\mathbf{x}\mathbf{W}_{k,1}^\top) = \mathbf{M}_k(\mathbf{x})\,(\mathbf{x}\mathbf{W}_{k,1}^\top) = (\mathbf{x}\mathbf{W}_{k,1}^\top)\,\mathbf{M}_k(\mathbf{x}). \tag{27}$$

The expert output becomes:

$$E_k(\mathbf{x}) = \mathbf{x}\,\mathbf{W}_{k,1}^\top\,\mathbf{M}_k(\mathbf{x})\,\mathbf{W}_{k,2}^\top = \mathbf{x}\,\big(\mathbf{W}_{k,2}\,\mathbf{M}_k(\mathbf{x})\,\mathbf{W}_{k,1}\big)^\top. \tag{28}$$

This is a convolution with an input-dependent effective kernel:

$$\mathbf{K}_k^{\text{eff}}(\mathbf{x}) = \mathbf{W}_{k,2}\,\mathbf{M}_k(\mathbf{x})\,\mathbf{W}_{k,1}. \tag{29}$$

The full MoE output:

$$\text{MoE}(\mathbf{x}) = \sum_{k=1}^{N} g_k(\mathbf{x})\, \mathbf{x}\,\big(\mathbf{K}_k^{\text{eff}}(\mathbf{x})\big)^\top = \mathbf{x}\Big(\textstyle\sum_{k=1}^{N} g_k(\mathbf{x})\,\mathbf{K}_k^{\text{eff}}(\mathbf{x})\Big)^\top. \tag{30}$$

This can be viewed as a dynamic convolution with $N$ input-dependent kernels, where both the routing weights $g_k(\mathbf{x})$ and the effective kernels $\mathbf{K}_k^{\text{eff}}(\mathbf{x})$ depend on the input.

**Interpretation.** The nonlinear case reveals that:

- Each expert implements a gated linear unit (GLU)-like computation with data-dependent masking.

- The MoE output is a weighted combination of masked convolutions.

- Both outer gating ($g_k$) and inner gating ($\mathbf{M}_k$) are input-dependent, creating a two-level conditional computation structure.

This analysis motivates designing cMoLLM with explicit control over both routing and activation patterns.

$\square$

# B. A Cluster-Structured Toy Model

We provide a simple toy model illustrating when MoE-style routing (and hence cMoLLM) is provably beneficial for cluster-structured data, in the spirit of Chen et al. (2022).

## B.1. Problem Setup

Consider a binary classification problem with input space $\mathbb{R}^d$ and two clusters per class. Let $\mu_{1,+}, \mu_{1,-}, \mu_{2,+}, \mu_{2,-} \in \mathbb{R}^d$ be four mean vectors, and let $\sigma^2 I$ be a shared covariance. We define the data distribution as

$$\mathbf{x} \mid y = +1 \sim \tfrac{1}{2}\mathcal{N}(\mu_{1,+}, \sigma^2 I) + \tfrac{1}{2}\mathcal{N}(\mu_{2,+}, \sigma^2 I), \tag{31}$$

$$\mathbf{x} \mid y = -1 \sim \tfrac{1}{2}\mathcal{N}(\mu_{1,-}, \sigma^2 I) + \tfrac{1}{2}\mathcal{N}(\mu_{2,-}, \sigma^2 I), \tag{32}$$

with prior $\mathbb{P}(y = +1) = \mathbb{P}(y = -1) = \tfrac{1}{2}$. We assume that clusters $(\mu_{1,+}, \mu_{1,-})$ and $(\mu_{2,+}, \mu_{2,-})$ are well separated and lie in different "regions" of the input space.

Formally, suppose there exists a unit vector $\mathbf{u} \in \mathbb{R}^d$ and scalars $a < b$ such that

$$\mathbf{u}^\top \mu_{1,+}, \mathbf{u}^\top \mu_{1,-} < a - \gamma, \tag{33}$$

$$\mathbf{u}^\top \mu_{2,+}, \mathbf{u}^\top \mu_{2,-} > b + \gamma, \tag{34}$$

for some margin $\gamma > 0$, and that the Bayes-optimal decision boundary within each cluster-pair is approximately linear in a (potentially different) direction.

## B.2. Expressivity of a Single Linear Classifier

Let $f_{\mathrm{lin}}(\mathbf{x}) = \mathrm{sign}(\mathbf{w}^\top \mathbf{x})$ be a linear classifier. Because the two class-conditional mixtures overlap across clusters, a single hyperplane must simultaneously separate both $(\mu_{1,+}, \mu_{1,-})$ and $(\mu_{2,+}, \mu_{2,-})$. When the optimal separating directions within the two regions are sufficiently misaligned, any single hyperplane incurs a non-negligible error.

The following statement summarizes this limitation at a high level.

**Proposition B.1** (Limitation of Single Linear Classifier). *Under the cluster separation conditions above, suppose that the optimal separating directions for the first and second cluster-pairs differ by an angle of at least $\theta_0 > 0$. Then there exists a constant $\varepsilon_0 = \varepsilon_0(\theta_0, \gamma, \sigma) > 0$ such that any linear classifier $f_{\mathrm{lin}}(\mathbf{x}) = \mathrm{sign}(\mathbf{w}^\top \mathbf{x})$ has misclassification error at least $\varepsilon_0$.*

The proof follows standard arguments for mixtures of Gaussians with incompatible linear separators and is omitted for brevity; see Chen et al. (2022).

## B.3. Two-Expert MoE/cMoLLM Construction

Now consider a two-expert MoE (or cMoLLM) model with a simple router that partitions space along direction $\mathbf{u}$:

$$g_1(\mathbf{x}) = \mathbb{I}(\mathbf{u}^\top \mathbf{x} \le \tau), \quad g_2(\mathbf{x}) = \mathbb{I}(\mathbf{u}^\top \mathbf{x} > \tau), \tag{35}$$

for some threshold $\tau \in (a, b)$. Let each expert be a linear classifier specialized to one region:

$$E_1(\mathbf{x}) = \mathrm{sign}(\mathbf{w}_1^\top \mathbf{x}), \tag{36}$$

$$E_2(\mathbf{x}) = \mathrm{sign}(\mathbf{w}_2^\top \mathbf{x}), \tag{37}$$

with $\mathbf{w}_1$ optimized for the first cluster-pair and $\mathbf{w}_2$ for the second. The overall prediction is

$$f_{\mathrm{moe}}(\mathbf{x}) = g_1(\mathbf{x})E_1(\mathbf{x}) + g_2(\mathbf{x})E_2(\mathbf{x}). \tag{38}$$

**Theorem B.2** (Toy Cluster Model: Benefit of Routing). *In the cluster-structured setting above, there exist parameters $(\mathbf{u}, \tau, \mathbf{w}_1, \mathbf{w}_2)$ such that the two-expert MoE (or cMoLLM) classifier $f_{\mathrm{moe}}$ attains misclassification error arbitrarily close to the Bayes-optimal error as the margin $\gamma$ increases and the covariance $\sigma^2$ decreases, while any single linear classifier $f_{\mathrm{lin}}$ suffers error at least $\varepsilon_0 > 0$ as in Theorem B.1.*

*Proof Sketch.* Because $\mathbf{u}$ separates the two cluster regions with margin $\gamma$, choosing $\tau \in (a, b)$ ensures that, with high probability (increasing as $\gamma/\sigma$ grows), samples from $(\mu_{1,+}, \mu_{1,-})$ fall into the first region and samples from $(\mu_{2,+}, \mu_{2,-})$ into the second. Within each region, the problem reduces to a two-component Gaussian mixture that is linearly separable by an appropriately chosen $\mathbf{w}_i$. Thus, the routed classifier $f_{\mathrm{moe}}$ can implement the Bayes-optimal decision rule up to an exponentially small error in $\gamma^2/\sigma^2$. On the other hand, Theorem B.1 implies that any single hyperplane must compromise between the two misaligned regions, incurring a constant error floor $\varepsilon_0$ even as $\gamma$ grows. Hence, for sufficiently well-separated clusters, the routed MoE/cMoLLM strictly outperforms any single linear classifier.

This toy example provides a simple, concrete setting where conditional computation (and by equivalence, dynamic convolution) is provably beneficial, aligning with the broader conclusions of Chen et al. (2022).

## C. Implementation Details

**Kernel Generator.** Each stream kernel $\mathbf{K}_k$ is initialized with Xavier initialization. We use no shared base or low-rank factorization in the main experiments.

**Gating Network.** The gating network is a single linear layer $\mathbf{W}_g \in \mathbb{R}^{d \times N}$. For `context_aware` gating, we add layer normalization before the projection. For `multi_head` gating, we use $H = 4$ heads with dimension $d/H$ each.

**Optimization.** We use AdamW with $\beta_1 = 0.9$, $\beta_2 = 0.95$, weight decay 0.1, and a cosine learning-rate schedule with linear warmup over the first 2000 steps.

## D. Hyperparameter Sensitivity

*Table 6.* Sensitivity to key hyperparameters (3 seeds; best value over mean $\pm$ std).

| Hyperparameter | Range | Best Value |
|---|---|---|
| Balance coefficient $\alpha$ | $\{0.001, 0.01, 0.1\}$ | 0.01 |
| Number of streams $N$ | $\{1, 2, 4, 8\}$ | 8 |

Table 6 summarizes the tested ranges and best values. cMoLLM is relatively robust to hyperparameter choices. The load-balancing coefficient $\alpha$ has the largest impact; values too small encourage stream collapse, while values too large hurt performance.

## E. Plain Language Summary

This paper introduces **cMoLLM**, a new way to scale large language models (LLMs) more efficiently. Traditional LLMs activate all parameters for every token, making training and inference expensive as models grow larger. Mixture-of-Experts (MoE) approaches try to address this by routing tokens to only a subset of "expert" networks, but most prior work applies this only to feed-forward layers and uses discrete Top-$K$ routing, which can be unstable. Our key insight is that MoE-style mixture layers can be exactly rewritten as dynamic convolutions: each expert corresponds to a convolution kernel, and the router mixes these kernels based on the input. We use this equivalence to design cMoLLM, which applies mixture routing to the entire LLM pipeline (not just feed-forward layers) using soft, fully differentiable gating—no Top-$K$, no virtual tokens, no auxiliary prediction branches. cMoLLM maintains a small set of parallel "streams," each with its own convolution kernel; a lightweight gating network produces input-dependent mixture weights, and the mixed kernel is applied via standard pointwise convolution, yielding parameter-efficient capacity scaling with stable training dynamics. On GPT-2–style models trained on FineWeb, cMoLLM improves perplexity and GLUE accuracy under matched compute, with better stream utilization and training stability than prior pipeline-level scaling methods like ParaScale and AltUp. The convolution-based design enables efficient implementation and favorable scaling in practice.

## F. Reproducibility Statement

Our implementation is based on PyTorch and follows standard Transformer architectures. The codebase will be made publicly available upon acceptance, including full model implementation (cMoLLM blocks, gating networks, training loop),

training scripts with hyperparameter configurations, evaluation scripts for language modeling and downstream tasks, and preprocessing scripts for the FineWeb dataset.

We use FineWeb (Penedo et al., 2024) for pretraining, which is publicly available. For downstream evaluation, we use standard benchmarks GLUE (Wang et al., 2019) and SQuAD (Rajpurkar et al., 2016), all of which are publicly available.

All hyperparameters are reported in Sections C, D and 5.1. Implementation details and hyperparameter sensitivity analysis are provided in Sections C and D. We use a GPT-2–style architecture (Radford et al., 2019) with 12 layers, hidden dimension 768, FFN dimension 3072, sequence length 4096, and learning rate $6 \times 10^{-5}$ with cosine schedule and 2000-step warmup. We use AdamW optimizer with $\beta_1 = 0.9$, $\beta_2 = 0.95$, and weight decay 0.1. For cMoLLM, we evaluate stream counts $N \in \{1, 2, 4, 8\}$; best validation loss in Table 2 is achieved at $n=8$ for `multi_head`. We use $N=8$ as a representative configuration in sensitivity and scaling tables. Load-balancing coefficient $\alpha = 0.01$.

Experiments were conducted on NVIDIA A100 GPUs. Training a single cMoLLM model ($N = 4$) for the reported experiments requires approximately 8 A100 GPU-days. Exact hardware specifications and software versions (PyTorch, CUDA, etc.) will be documented in the code repository.

Language modeling metrics (loss, perplexity) are computed on validation splits. Downstream task evaluation follows standard protocols: GLUE use development set accuracy; SQuAD uses F1 score. All experimental results use **3 random seeds** and are reported as **mean $\pm$ standard deviation (std)** throughout the paper (main tables, scaling tables, and appendix). A plain language summary is provided in Section E.

We compare against a dense GPT-2 baseline (standard Transformer), ParaScale (Chen et al., 2025) (reimplemented with virtual token streams), and AltUp (Baykal et al., 2023) (reimplemented with auxiliary prediction branch). All baselines use identical training data, hyperparameters where applicable, and evaluation protocols to ensure fair comparison. Reproducibility details are provided in Section F.

## G. Use of LLM

The authors used generative AI tools (Grammarly, ChatGPT) only for grammar checking and language polishing. All technical content, experimental design, data analysis, and conclusions were generated and verified by the human authors. The use of AI tools does not affect the originality or authorship of this work.

