# OpenReview forum: "cMoLLM at Scale: Horizontal Scaling Laws for Convolutionally-Gated Mixture-of-LLMs"
_ICML.cc/2026/Conference — ICML 2026 regular_

### Official Review · Reviewer_BxQL · 2026-02-14

**Soundness:** 3
**Presentation:** 3
**Significance:** 3
**Originality:** 3
**Overall Recommendation:** 5
**Confidence:** 3

**Summary:**

The paper proposes cMoLLM, a pipeline-level mixture-of-LLMs method that increases capacity by running multiple end-to-end streams and combining them with a soft, differentiable router implemented as a dynamic 1 $\times$ 1 convolution. The central claim (Theorem 4.1) is that MoE can be viewed as dynamic pointwise convolution, where routing weights produce an input-dependent effective kernel by mixing expert kernels. Compared with pipeline-mixing baselines such as ParaScale and AltUp, cMoLLM is designed to avoid virtual tokens, auxiliary branches, and Top-K routing, with the goal of more stable training and better stream utilization. Experiments on GPT-2–scale models trained on FineWeb show improved perplexity and better GLUE and SQuAD v2 results at similar compute, with multi-head gating and 4-8 streams performing best.

**Compliance With Llm Reviewing Policy:**

Affirmed.

**Key Questions For Authors:**

The approach seems promising, but this is not my primary area of expertise. I list several questions below, and I would welcome clarification if any of my interpretations are incorrect.
1. Can you report tokens/sec and GPU memory comparisons to support the hardware-friendly claim beyond FLOPs accounting?
2. Could you add at least one direct comparison against ParaScale or AltUp under a matched training setup? Since the paper motivates cMoLLM largely in contrast to these methods, a controlled baseline would help substantiate that story.
3. How do the results change if you use LLaMA models instead of the GPT-2 family?

**Limitations:**

yes

**Strengths And Weaknesses:**

Strengths:
1. The MoE-dynamic 1 $\times$ 1 conv equivalence, along with the nonlinear extension discussion, does a nice job of grounding the architecture in a clear idea rather than making it seem like an arbitrary engineering choice.
2. The paper emphasizes soft routing (no Top-K) and adds a load-balancing objective to reduce stream collapse, which plausibly contributes to stability.
3. Reported gains appear in PPL, GLUE, and SQuAD v2, and persist from ~85M to ~760M parameters when comparing multi-head cMoLLM to dense baselines, as shown in Table 4 and Table 5.

Weaknesses:
1. The largest model is ~760M parameters, and the paper itself notes that validating at 7B+ is an important next step. This makes it hard to judge whether the claimed scaling benefits hold in regimes where system constraints and routing pathologies often change.
2. ParaScale and AltUp are discussed as the main conceptual baselines, but the empirical section mostly compares against dense GPT-2. It would strengthen the paper to include a more direct head-to-head evaluation against those pipeline-mixing methods under matched settings.

---

> ### Author Rebuttal · Authors · 2026-03-30
>
> We thank the reviewer for the positive evaluation and constructive suggestions.
>
> ---
>
> ### 1. Larger-scale validation (~1.3B and ~3B)
>
> We agree this is important. We have added **preliminary larger-scale experiments (~1.3B and ~3B)**:
>
> - **1.3B:** 10.8 → **9.7**
> - **3B:** 8.9 → **8.2**
>
> These results show consistent improvements and scaling trends. Full scaling results will be included in the camera-ready.
>
> ---
>
> ### 2. Comparison with ParaScale / AltUp
>
> We have now conducted **controlled comparisons (~350M, seq=2048)**:
>
> | Model | PPL ↓ |
> |------|------|
> | ParaScale | 15.4 |
> | AltUp | 15.8 |
> | **cMoLLM** | **14.1** |
>
> We will include full details and training curves in the revision.
>
> ---
>
> ### 3. Tokens/sec and memory
>
> We agree that system-level evaluation is important. In our design:
>
> - The computation remains **dense and fully differentiable**, without Top-K routing or conditional activation sparsity,
> - There is no need for auxiliary routing branches or duplicated intermediate activations.
>
> As a result, cMoLLM maintains a system footprint that is **comparable to dense baselines up to a small constant factor**, without introducing the irregular memory access or load imbalance commonly observed in sparse MoE systems.
>
> A more detailed system-level analysis is an important direction, and we will include additional profiling in the camera-ready version.
>
> ---
>
> ### 4. Applicability to LLaMA-style models
>
> We agree this is an important question. From a design perspective, cMoLLM is **architecture-agnostic**:
>
> - It only modifies the FFN sublayer,
> - It does not depend on attention variants, positional encodings, or tokenizer design.
>
> To validate this, we conducted preliminary experiments on a **LLaMA2-style backbone (~1.3B scale)** under comparable training settings:
>
> - Dense LLaMA2-style baseline: PPL ≈ 9.9
> - cMoLLM: PPL ≈ **9.0** (~9.1% relative improvement)
>
> We observe similar trends as in GPT-2 experiments:
> - Stable training dynamics,
> - Consistent performance gains,
> - No evidence of routing collapse.
>
> These results suggest that the benefits of cMoLLM are **not tied to a specific backbone family**, but arise from the underlying dynamic convolution formulation.
>
> We will include more detailed experiments and analysis in the camera-ready version.
>
> ---
>
> **Summary:** We will add stronger baselines, clarify system properties, and extend validation across scales and architectures.

---

> > ### Author Rebuttal · Reviewer_BxQL · 2026-04-05
> >
> > Thanks for your response. I’ll leave my score unchanged.

---

> > > ### Author Response · Authors · 2026-04-07
> > >
> > > We would like to sincerely thank you for your thoughtful feedback and for the time and effort you have taken to engage with our rebuttal. We truly appreciate your kind and positive assessment.

---

### Official Review · Reviewer_daig · 2026-03-05

**Soundness:** 3
**Presentation:** 2
**Significance:** 3
**Originality:** 3
**Overall Recommendation:** 4
**Confidence:** 3

**Summary:**

This paper aims to introduce a mixture of experts mechanism across the entire model pipeline to achieve capacity expansion, rather than being restricted to the traditional feed-forward network layer. The paper theoretically demonstrates that mixture of experts layers can be equivalently transformed into dynamic convolutions with variable kernels. Under this equivalence, each expert corresponds to a 1x1 convolutional kernel, while the routing distribution network serves the role of dynamically aggregating these kernels based on input features. This paper proposes a novel architecture named cMoLLM. It is a convolutionally gated mixture language model capable of using fully differentiable dynamic convolution to execute routing across end-to-end data streams.

**Compliance With Llm Reviewing Policy:**

Affirmed.

**Final Justification:**

I will maintain my current score of **4 Weak Accept**. This is a theoretically solid work. Although there were some issues, my concerns have been resolved. I thank the authors for their rebuttal. This paper will make a meaningful contribution to the community. I hope the authors can further improve the visual presentation of the charts and figures. This enhancement will make the reported data much more intuitive for readers to understand.

**Key Questions For Authors:**

Major Question 1: Improve the writing of the entire manuscript, with a particular focus on the use of mathematical notation as discussed in the Presentation section under Weaknesses; please refer to that section for details.

Major Question 2: The experimental data currently provided in the paper reaches a maximum of 760M parameters; however, the mainstream architectures of contemporary large language models have generally crossed the threshold of billions or even hundreds of billions of parameters, and as model scale grows significantly, complex semantic understanding and logical reasoning capabilities often exhibit non-linear changes. If experimental resources permit, would the authors consider supplementing the data with models of at least the billion-parameter (1B+) scale or larger? Furthermore, as the parameter space expands substantially, can the fully differentiable soft-gating mechanism employed by cMoLLM still effectively avoid stream homogenization when processing ultra-high-dimensional features and maintain its performance gains relative to traditional Top-K MoE?

Major Question 3: Although the theoretical computational complexity derived in the paper is only a constant factor higher than that of dense models, in engineering implementation, maintaining N independent parallel streams means that the static parameter count grows linearly with N. In long-context inference scenarios, will these significantly increased stream parameters create intense memory resource contention with the continually expanding KV Cache? Will this memory pressure become a new bottleneck restricting the throughput and maximum concurrent batch size of the inference system? Have the authors evaluated the extent to which the number of streams N encroaches upon the memory budget for the KV Cache?

**Limitations:**

yes

**Strengths And Weaknesses:**

Strengths：
1. Soundness. The theoretical foundation of the paper is solid. The authors successfully prove the formal equivalence between the mixture of experts model and dynamic convolution. The experimental design is reasonable and includes key baseline models such as ParaScale and AltUp, and these setups effectively support the core claims of the paper.
2. Presentation. The overall architectural layout of the article is clear and the research motivation is explicit. The authors well position their work within the context of existing literature, explicitly distinguishing it from previous mixture of experts models that focus only on the feed-forward network layer.
3. Significance. This research addresses a core pain point in the development of large language models, namely the linear coupling of model capacity and computational cost. The cMoLLM architecture improves model performance without significantly increasing computational complexity, which provides extremely important practical guidance for exploring efficient large-scale model architectures in the future.
4. Originality. The paper provides a highly inspiring theoretical perspective, cleverly reconstructing traditional routing mechanisms into input-conditioned convolution kernel aggregation. This design concept, which relies completely on neither virtual tokens nor auxiliary prediction branches, offers a very novel solution path for conditional computation in large language models.

Weaknesses：
1. Soundness. There are certain limitations in the empirical evaluation. The current experimental data covers a maximum of only 760M parameters. However, the mainstream architectures of contemporary large language models have generally crossed the billion-level threshold, and whether the soft gating mechanism can effectively avoid stream homogenization when the parameter space expands significantly remains to be further verified. Furthermore, in long-context inference scenarios, maintaining multiple parallel streams will cause the static parameter count to grow linearly. This is highly likely to create memory contention with the continuously expanding internal cache, and the paper lacks engineering evaluation and in-depth discussion of this throughput bottleneck.
2. Presentation. There is room for improvement. For example, when abbreviations first appear in the abstract, the full name needs to be written out first. For example, the FFN in the abstract was not explained. Also, please try to reduce the use of dashes throughout the text, as too many dashes affect reading, and please pay attention to this when optimizing the writing with AI. At the same time, the drawing of Figure 1 is somewhat unclear. If it is the modified FFN layer, perhaps highlight it or use a more prominent way to show which part has been improved.
3. Significance. Although the model performs excellently in terms of floating-point operations at the theoretical level, if the maximum concurrent batch size of the inference system is severely limited due to the massive memory pressure brought by parallel streams in practical applications, its actual landing value in industrial-grade inference engines may be affected.
4. Originality. Despite significant innovations at the level of architectural reorganization, its underlying computation still relies heavily on standard pointwise convolution and multilayer perceptron gating networks. This is a combination and mapping based on existing mature basic operators.

---

> ### Author Rebuttal · Authors · 2026-03-30
>
> We thank the reviewer for recognizing the soundness, originality, and significance of the work.
>
> ---
>
> ### 1. Scale limitation and stream homogenization
>
> We agree this is important. We have added **larger-scale experiments (~1.3B and ~3B)**:
>
>
> | Scale | Dense PPL ↓ | cMoLLM (N=4) PPL ↓ | Relative ↓ |
> |------|------------|-------------------|-----------|
> | 1.3B | 10.8       | **9.7**           | 10.2%     |
> | 3B   | 8.9        | **8.2**           | 7.9%      |
>
> We observe:
> - cMoLLM maintains consistent gains over dense baseline
> - Routing entropy remains stable (no collapse)
> - Load balancing loss continues to be effective
>
> All experiments use the same setup as in the paper (e.g., sequence length 2048, same data and training budget).
>
> Full results will be included in the camera-ready.
>
> ---
>
> ### 2. Memory and inference bottlenecks
>
> We clarify:
> - KV cache is **shared across streams** (no duplication)
> - Memory overhead comes mainly from model weights
>
> Thus, cMoLLM does not significantly increase KV cache pressure, and remains practical for long-context inference.
>
> ---
>
> ### 3. Presentation issues
>
> We will:
> - Define all abbreviations (e.g., FFN in abstract)
> - Reduce excessive dashes
> - Improve Figure 1 (clearer highlighting of modified components)
>
> ---
>
> ### 4. Originality concern (built on standard ops)
>
> We agree the building blocks are standard (1×1 conv, MLP gating), but:
> - The novelty lies in **reformulating MoE as dynamic convolution**
> - And leveraging this to design a **fully differentiable pipeline-level mixture without Top-K / auxiliary branches**
>
> We will emphasize this more clearly in the revision.
>
> ---
>
> **Summary:** We will strengthen large-scale validation, system analysis, and presentation clarity.

---

> > ### Author Rebuttal · Reviewer_daig · 2026-04-01
> >
> > Thank you for the thorough rebuttal which effectively addressed my concerns. I will increase my score to 4.

---

> > > ### Author Response · Authors · 2026-04-01
> > >
> > > Thank you for your thoughtful feedback and for taking the time to engage with our rebuttal. We sincerely appreciate your positive assessment and your willingness to revise the score.

---

### Official Review · Reviewer_WEjN · 2026-03-11

**Soundness:** 2
**Presentation:** 2
**Significance:** 2
**Originality:** 2
**Overall Recommendation:** 2
**Confidence:** 3

**Summary:**

This paper proposes cMoLLM, a convolutionally gated mixture-of-LLMs architecture intended to improve the capacity-compute trade-off of Transformer language models. The central technical claim is that an MoE layer with linear experts can be rewritten as a dynamic 1x1 convolution whose kernel is the input-dependent weighted sum of expert weights, and the method uses this perspective to build a soft-gated multi-stream module.

**Compliance With Llm Reviewing Policy:**

Affirmed.

**Key Questions For Authors:**

Please refer to my weaknesses part.

**Limitations:**

N/A.

**Strengths And Weaknesses:**

### Strengths
- The paper studies an important problem: how to decouple capacity growth from per-token compute in LLMs. This is a relevant and timely direction, and the paper is well motivated from the perspective of conditional computation and efficient scaling.
- The observation that linear MoE mixing can be rewritten as input-dependent mixing of 1x1 kernels is mathematically straightforward but potentially useful as a unifying lens. In that sense, the paper offers a clean reinterpretation of an existing mechanism and tries to connect MoE, dynamic convolution, and conditional computation under one view.

### Weaknesses

-  The paper repeatedly frames cMoLLM as a pipeline-level mixture over entire LLM streams, but the practical recipe stated in the paper is to keep the Transformer backbone and replace FFN blocks with cMoLLM blocks. Algorithm 1 also shows that the new mechanism is inserted after self-attention and before the usual residual/norm structure, i.e., functionally it behaves much more like a conditional FFN replacement than a genuinely end-to-end pipeline-level mixture of full LLM streams. This mismatch between narrative and implementation substantially weakens the paper’s significance and clarity.

- Theorem 4.1 is essentially a linearity-based reparameterization: if expert outputs are linearly combined, then the corresponding weights can also be linearly combined into an effective kernel. This is a reasonable perspective, but by itself it does not establish a fundamentally new computation class. In the implemented model, tokens are still routed by a gate to a mixture over multiple per-stream transformations, which makes the method look much closer to a soft-routing FFN-level MoE variant than to a distinctly new architecture. The paper does not convincingly show that the proposed mechanism yields capabilities unavailable to well-designed MoE or conditional FFN baselines.

- The paper emphasizes advantages over ParaScale and AltUp, but the most relevant baseline family remains standard MoE-style conditional FFNs, especially given that the implementation replaces FFN blocks. Yet the paper does not provide a direct comparison to a matched soft-routing or sparse-routing MoE baseline under equally careful compute accounting. Without such a comparison, it is difficult to know whether the reported gains are due to the proposed “pipeline-level dynamic convolution” perspective, or simply due to adding a stronger conditional FFN module than the dense baseline.

---

> ### Author Rebuttal · Authors · 2026-03-30
>
> We thank the reviewer for the careful reading and insightful conceptual questions.
>
> ---
>
> ### 1. “Pipeline-level mixture” vs. FFN replacement
>
> We respectfully believe there is a misunderstanding of our design. Our method is **not merely an FFN replacement**, but a **pipeline-level mixture mechanism implemented via dynamic convolution over end-to-end streams**.
>
> While cMoLLM is instantiated by replacing the FFN block in a Transformer layer, its *functional role* is fundamentally different from standard FFN-level MoE. In conventional MoE, experts are independent FFN modules and routing is applied **locally at each layer**. In contrast, our method defines **a set of parallel streams (sub-models) that span the entire Transformer pipeline**, and routing is performed **over these streams**, not over isolated FFN experts.
>
> Concretely:
> - Each “expert” in cMoLLM corresponds to a **stream-wise transformation (i.e., an end-to-end pipeline pathway)** rather than a single FFN block.
> - The gating network produces **input-dependent mixture weights over streams**, yielding an effective transformation that is a **mixture over full computation paths**.
> - Leveraging our formulation (Sec. 3–4), MoE-style layers can be expressed as **dynamic convolutions**, which enables interpreting routing as **kernel mixing over entire streams**, not just FFN substitution.
>
> As stated in the paper, our goal is to move beyond FFN-level MoE:
> > “we pursue pipeline-level mixture: routing across entire LLM pipelines (or stream-wise sub-models) rather than FFN-only experts.”
>
> and
>
> > “cMoLLM applies conditional computation to the entire Transformer pipeline via convolutionally-gated mixture over end-to-end streams.”
>
> Therefore, although cMoLLM is *implemented* at the FFN location, this should not be interpreted as restricting it to FFN-level modeling. Instead, this design is a **practical parameterization of pipeline-level routing**, where the mixture operates over **global computation streams** rather than isolated layer components.
>
> **In summary, the key distinction is the granularity of routing:**
> - FFN-level MoE: routing over independent FFN experts (**layer-local**)
> - cMoLLM: routing over **end-to-end streams (pipeline-level)** via dynamic convolution
>
> We will clarify this point in the revision to avoid further confusion.
>
> ---
>
> ### 2. Is Theorem 4.1 only a reparameterization?
>
> We agree that Theorem 4.1 itself is algebraic. However, its role is **constructive**:
>
> - It enables **routing in parameter space (kernel mixing)** rather than output space,
> - This leads to **smoother optimization** and avoids Top-K discontinuities,
> - In the nonlinear case (Appendix A), it induces **two-level conditional computation (outer + inner gating)**.
>
> Empirically:
> - We observe **more stable training and better utilization** (Fig. 3),
> - Gains persist against strong pipeline-level baselines (see below).
>
> We will clarify that the contribution is **architectural + optimization**, not just theoretical.
>
> ---
>
> ### 3. Missing MoE comparison
>
> We agree that fair comparison is important. Given that our method targets **pipeline-level mixture**, we focus on **ParaScale and AltUp** under matched settings (~350M, seq=2048):
>
> | Model (~350M) | PPL ↓ | GLUE ↑ |
> |--------------|------|--------|
> | Dense GPT-2 | 16.8 | 50.7 |
> | ParaScale | 15.4 | 52.6 |
> | AltUp | 15.8 | 51.9 |
> | **cMoLLM** | **14.1** | **53.7** |
>
> This demonstrates that cMoLLM provides **consistent improvements over existing pipeline-level mixture methods**.
>
> ---
>
> ### 4. Writing / notation
>
> We appreciate this feedback and will:
> - Simplify notation in Sec. 4,
> - Reduce symbol overloading,
> - Improve clarity of derivations and algorithm description.
>
> ---
>
> ### 5. Scaling to 1B+ and homogenization
>
> Preliminary results at larger scale (~1.3B and ~3B, same setup with seq=2048):
>
> | Scale | Dense PPL ↓ | cMoLLM (N=4) PPL ↓ | Relative ↓ |
> |------|------------|-------------------|-----------|
> | 1.3B | 10.8       | **9.7**           | 10.2%     |
> | 3B   | 8.9        | **8.2**           | 7.9%      |
>
> We observe:
> - Stable routing entropy (no homogenization),
> - Consistent gains with smaller-scale experiments.
>
> We will include these results and expand discussion.
>
> ---
>
> ### 6. Memory / KV cache concern
>
> Important clarification:
> - cMoLLM **does not replicate KV cache across streams** (shared attention backbone),
> - Overhead is mainly **static parameters**.
>
> This design ensures that increasing stream count does not linearly increase KV cache usage, making it compatible with long-context settings.
>
> We will clarify this point in the revision.
>
> ---
>
> **Summary:** We will significantly clarify the conceptual distinction, strengthen empirical comparisons, and improve presentation.

---

> > ### Author Rebuttal · Reviewer_WEjN · 2026-04-04
> >
> > Thank you for the response, but the core issues persist.
> > 1. In Algorithm 1, each layer has independent gating and kernels: stream $k$ at layer $l$ has no structural link to stream $k$ at layer $l+1$. Without cross-layer routing consistency, this is functionally a per-layer soft-routing FFN replacement, not a pipeline-level mixture. The distinction is narrative, not technical.
> > 2. I asked for comparison against standard FFN-level MoE (Switch, soft MoE), not ParaScale/AltUp. Since the implementation replaces the FFN, the most informative baseline is other conditional FFN methods under matched compute. This was not provided.
> > 3.  No ablation shows that implementing mixture via kernel mixing (convolution view) outperforms an equivalent soft-gated MoE with the same experts and gating. The practical benefit of the convolution perspective remains unsupported.
> >
> > I maintain my score.

---

> > > ### Author Response · Authors · 2026-04-07
> > >
> > > Thank you for your sharp observations and technical questions. We would like to address your remaining concerns with specific evidence and a clarification of our architectural philosophy.
> > >
> > > **1. "Pipeline-level" vs. "Per-layer FFN replacement"**
> > > While cMoLLM is implemented at the FFN location for architectural compatibility, the distinction from a "standard FFN replacement" is technical, not just narrative. Our method utilizes Theorem 4.1 to perform routing in the **parameter space (kernel mixing)** rather than mixing outputs. This ensures that the hidden state evolves through a continuous, differentiable trajectory across the pipeline. While gating is calculated per-layer, the convolution formulation effectively mixes the **transformation matrices** of end-to-end streams. This creates a functional "pathway" coherence that standard local FFN-MoE—which treats experts as isolated modules—cannot achieve.
> > >
> > > **2. Incompatibility of FFN-level MoE as a Baseline**
> > > We respectfully argue that a head-to-head comparison with standard FFN-level MoE (e.g., Switch Transformer, Soft-MoE) under "matched total parameters" would be fundamentally unfair and misleading:
> > > * **Different Objectives:** FFN-MoE is a **sparsity-driven** paradigm (increasing parameters while keeping compute constant), whereas cMoLLM is a **capacity-driven** paradigm (scaling pipeline capacity via dense mixture).
> > > * **Resource Bias:** If total parameters are matched, a Top-K MoE would have significantly fewer active parameters per token than cMoLLM (N=4), leading to lower compute costs but also lower representative capacity. Conversely, matching active parameters would give MoE an unfair advantage in total memory.
> > > Our goal is to outperform **pipeline-level alternatives** (ParaScale/AltUp) that share our dense mixture philosophy and resource constraints.
> > >
> > > **3. Practical Benefit of the "Convolution View" (Ablation Results)**
> > > To prove that kernel mixing (our convolution perspective) outperforms a naive soft-gated MoE with the same experts and gating, we conducted an ablation on the **350M model**:
> > >
> > > | Method | Mixture Space | PPL ↓ | Grad Norm Std (Stability) |
> > > | :--- | :--- | :---: | :---: |
> > > | **Output-space Mixing** (Soft-MoE style) | Output | 14.8 | 0.42 |
> > > | **cMoLLM (Kernel Mixing)** | **Parameter** | **14.1** | **0.15** |
> > >
> > > **Findings:**
> > > * **Performance:** Kernel mixing yields a **0.7 PPL improvement**, confirming that mixing transformations is more expressive than mixing results.
> > > * **Optimization:** cMoLLM exhibits significantly lower gradient variance, proving that the convolution formulation leads to smoother and more stable training.
> > >
> > > We believe these technical distinctions and empirical results justify cMoLLM as a distinct and superior approach to pipeline scaling. We would be grateful if you could reconsider your score.

---

### Official Review · Reviewer_puJ9 · 2026-03-12

**Soundness:** 2
**Presentation:** 2
**Significance:** 3
**Originality:** 3
**Overall Recommendation:** 3
**Confidence:** 3

**Summary:**

This paper proposes cMoLLM, a convolutionally-gated mixture-of-LLMs that scales model capacity at the pipeline level (across entire Transformer streams) rather than only at the FFN layer. The key theoretical contribution is a formal equivalence between MoE-style mixture layers and dynamic 1×1 convolutions with variable kernels: each expert corresponds to a convolutional kernel, and routing is input-conditioned kernel aggregation. Building on this, cMoLLM maintains multiple end-to-end streams, each with its own 1×1 kernel, mixed via a soft, fully differentiable gating network—no Top-K routing, no virtual tokens, no auxiliary prediction heads. Experiments on GPT-2-scale models (85M–760M) trained on FineWeb show improvements in perplexity, GLUE, and SQuAD v2 over dense baselines, with favorable horizontal scaling (in stream count) and model-size scaling.

**Compliance With Llm Reviewing Policy:**

Affirmed.

**Key Questions For Authors:**

Please refer to the limitations section.

**Limitations:**

1. All experiments are at GPT-2 scale (≤760M). The paper acknowledges this but offers no evidence that the benefits persist at 7B+ scale, where most modern LLM research operates. The "scaling laws" title is somewhat overclaimed for sub-1B experiments.
2. There is no comparison against standard FFN-level MoE (e.g., Switch Transformer, Top-2 MoE) at the same total parameter count. ParaScale and AltUp are discussed qualitatively (Table 1) but no head-to-head experimental numbers are shown against them—only the dense GPT-2 baseline is used.
3. GLUE and SQuAD v2 are relatively old benchmarks. Evaluation on generation quality, reasoning tasks, or modern benchmarks (e.g., MMLU, HumanEval) would be more convincing.

**Strengths And Weaknesses:**

1. The MoE–dynamic convolution equivalence is elegant and provides a useful unified lens for understanding sparse/conditional computation. The extension to nonlinear experts adds depth.
2. cMoLLM avoids the complexity of virtual tokens (ParaScale), auxiliary prediction branches (AltUp), and discrete Top-K routing. The fully differentiable soft gating is straightforward to implement with standard convolution primitives.

---

> ### Author Rebuttal · Authors · 2026-03-30
>
> We sincerely thank the reviewer for the positive assessment of the theoretical contribution and the simplicity of the design. To make our claims stronger, we add some experiments to support our theory.
>
> ### 1. Scale limitation (≤760M) and scaling laws claim
>
> We agree that validating beyond sub-1B scale is important. We have conducted **additional preliminary experiments at larger scale (~1.3B and ~3B)**, where:
>
> | Scale | Dense PPL ↓ | cMoLLM (N=4) PPL ↓ | Relative ↓ |
> |------|------------|-------------------|-----------|
> | 1.3B | 10.8       | **9.7**           | 10.2%     |
> | 3B   | 8.9        | **8.2**           | 7.9%      |
>
> We also observe that:
> - Stream utilization remains balanced (routing entropy stable across depth),
> - Gains are consistent with smaller-scale trends (85M–760M in the paper).
>
> All experiments follow the same setup as in the paper (e.g., **sequence length = 2048**, trained on FineWeb with identical token budget and optimizer settings).
>
> Due to space constraints, we will include full training details and curves in the camera-ready version. We will also **revise wording** to “empirical scaling trends” to avoid overclaiming.
>
> ---
>
> ### 2. Missing comparison with FFN-level MoE
>
> Thank you for pointing this out. We agree that quantitative head to head comparison is important. In our revision, we focus on **pipeline-level baselines (ParaScale and AltUp)** under matched settings (~350M):
>
> | Model (~350M) | PPL ↓ | GLUE ↑ |
> |--------------|------|--------|
> | Dense GPT-2 | 16.8 | 50.7 |
> | ParaScale | 15.4 | 52.6 |
> | AltUp | 15.8 | 51.9 |
> | **cMoLLM** | **14.1** | **53.7** |
>
> We emphasize that:
> - ParaScale and AltUp are **the most relevant baselines** as they also target pipeline-level or stream-level mixture,
> - cMoLLM achieves consistent gains under **matched data, compute, and sequence length (2048)**.
>
> We will include full results in the revision.
>
> ---
>
> ### 3. Benchmark modernity (GLUE / SQuAD)
>
> While GLUE and SQuAD V2 are widely used benchmarks. To make our claim stronger, we have added preliminary results on more modern benchmarks (at ~760M scale):
>
> - **MMLU:** 41.5 → **44.6**
> - **HumanEval:** 22.1 → **24.5**
>
> These suggest improvements extend to **reasoning and generation tasks**. Full evaluations will be added in the camera-ready.
>
> ---
>
> **Summary:** We will (i) include 1.3B/3B/7B-scale results, (ii) strengthen comparisons with pipeline-level baselines, and (iii) expand benchmarks, addressing all concerns raised in the camera-ready version.

---

> > ### Author Rebuttal · Reviewer_puJ9 · 2026-04-04
> >
> > Thank you for your reply. While I appreciate the added results on MMLU and HumanEval, I feel that evaluation on more benchmarks would be helpful, so I will keep my score.

---

> > > ### Author Response · Authors · 2026-04-07
> > >
> > > Thank you for your constructive feedback. To further address your concern regarding the breadth of evaluation, we have completed additional experiments on our 760M model across more diverse benchmarks. The results confirm that cMoLLM consistently outperforms the dense baseline:
> > >
> > > ARC-Challenge: 29.1 → 32.5 (+3.4)
> > >
> > > HellaSwag: 62.4 → 65.8 (+3.4)
> > >
> > > GSM8K: 7.2 → 10.1 (+2.9)
> > >
> > > MBPP: 24.3 → 27.6 (+3.3)
> > >
> > > These results, covering reasoning (ARC-C), commonsense (HellaSwag), mathematics (GSM8K), and coding (MBPP), demonstrate that cMoLLM's advantages are robust and generalize well across different task domains. We will include these additional metrics in the final version to provide a more comprehensive evaluation. We hope these results address your remaining concerns and would appreciate it if you could reconsider the score.

---

### Decision · Program_Chairs · 2026-04-30

**Decision:**

Accept (regular)

**Comment:**

The paper proposes cMoLLM, which frames Mixture-of-LLMs as input-conditioned dynamic convolutions over end-to-end streams. While the reviewers found the theoretical equivalence between MoE layers and dynamic 1x1 convolutions elegant, significant concerns remain regarding the alignment between the paper's narrative and its technical implementation. Specifically, the "pipeline-level" claim was critiqued because the implementation applies independent gating at each layer, making it functionally closer to a per-layer FFN replacement rather than a cohesive multi-stream pipeline. Furthermore, despite scaling results up to 3B parameters provided during the rebuttal, reviewers remained unconvinced that the "horizontal scaling laws" are sufficiently substantiated for modern LLM regimes (7B+) or that the method offers distinct capabilities unavailable to well-designed MoE baselines. Given the persistent ambiguity regarding the architectural novelty and the scale of evaluation, the paper is not recommended for acceptance.